# The Cat Mandible (I): Anatomical Basis to Avoid Iatrogenic Damage in Veterinary Clinical Practice

**DOI:** 10.3390/ani11020405

**Published:** 2021-02-05

**Authors:** Matilde Lombardero, Diana Alonso-Peñarando, María del Mar Yllera

**Affiliations:** 1Unit of Veterinary Anatomy and Embryology, Department of Anatomy, Animal Production and Clinical Veterinary Sciences, Faculty of Veterinary Sciences, University of Santiago de Compostela–Campus of Lugo, 27002 Lugo, Spain; mar.yllera@usc.es; 2Department of Animal Pathology, Faculty of Veterinary Sciences, University of Santiago de Compostela–Campus of Lugo, 27002 Lugo, Spain; diana.alonso.penarando@rai.usc.es; 3DVM at Veterinary Clinic Villaluenga, Villaluenga de la Sagra, 45520 Toledo, Spain

**Keywords:** anatomy, feline, lower jaw, neurovascular supply, temporomandibular joint, tooth

## Abstract

**Simple Summary:**

Nowadays, cats are one of the most common companion animals. They differ from dogs in some important aspects. However, most of the veterinary clinics are oriented towards the care and treatment of dogs, where the cat patient is clinically treated like a small dog. The cat mandible and related structures have some particularities that should be taken into account, when treating a cat, to avoid any unintended medical (iatrogenic) damage. The feline mandible has fewer teeth than a dog’s one, but tooth roots and the neurovascular supply account for up 70% of the volume of the mandibular body. This fact makes mandibular fracture repair challenging. In addition, the cat mandible has a prominent angular process that, when the cat is under anesthesia and his mouth is wide open (during oral or transoral manipulation), compresses the maxillary artery (that supplies blood to the brain) inducing temporal or permanent blindness and/or deafness. Other particularities of the cat jaw are also addressed to get a comprehensive knowledge of its functional anatomy, essential to an effective feline clinical practice.

**Abstract:**

Cats are one of our favourite pets in the home. They differ considerably from dogs but are usually treated clinically as small dogs, despite some anatomical and physiological dissimilarities. Their mandible is small and has some peculiarities relative to the dentition (only three incisors, a prominent canine, two premolars and one molar); a conical and horizontally oriented condyle, and a protudent angular process in its ventrocaudal part. Most of the body of the mandible is occupied by the mandibular dental roots and the mandibular canal that protects the neurovascular supply: the inferior alveolar artery and vein, and the inferior alveolar nerve that exits the mandible rostrally as the mental nerves. They irrigate and innervate all the teeth and associated structures such as the lips and gingiva. Tooth roots and the mandibular canal account for up to 70% of the volume of the mandibular body. Consequently, when fractured it is difficult to repair without invading the dental roots or vascular structures. Gaining a comprehensive anatomical knowledge and good clinical practice (such as image diagnosis before and post-surgery) will help in the awareness and avoidance of iatrogenic complications in day-to-day feline clinical practice.

## 1. Introduction

Cats have been fully appreciated animals for a long time. Ancient Egyptians mummified cats, as pets were buried with their owner, or they were used as votive offerings that depicted the gods [1]. Animals were revered by associating them with deities. In fact, the goddess Bastet was symbolised as a cat or even a woman with a feline head [2].

Nowadays, cats are one of the most numerous pets at home, sharing our company and friendship. According to the American Pet Products Association’s (APPA) 2019–2020 National Pets Owners Survey [3], it is estimated that in the USA, 67% of households own a pet; that is 84.9 million homes with an estimated 63.4 million dogs and 42.7 million cats, among other pets (such as freshwater/saltwater fish, birds, small animals and reptiles, accounting all together for 137 million animals being kept at home). However, compared with 2019 figures, in the USA, an estimated 89.7 million dogs and 94.2 million cats is reported (made in base of multiplying the number of households that own the pet by the average number of pets owned by household), which is a large decrease in both species, but no explanation was given, unless maybe it was a poor rough calculation. In Europe, there were at least 85 million households (38% of all households) owning a minimum of one pet animal [4]: 106.4 million cats and 77.4 million dogs. Hence, in Europe, the numbers of cats largely overtake those of dogs as pets; as they do not take much space, they do not need to go outdoors during the day, and their ownership is more affordable, among other reasons. However, the majority of pet clinics are more oriented towards dog management and treatment. Concerning cat size, they are usually small and consequently their surgery is more difficult. In addition, their lifespan is longer than the dog’s, with an average life span of 12 years and ages of 20 years or more not uncommon. The current longevity record cited by Beaver in 2003 [5] was 36 years. Regarding clinical surgery solutions, up to now, little has been is done by companies to provide surgical prostheses and other products adapted to such a small animal, as they are more oriented towards different dog breeds. Hence, unfortunately, and in accordance with Little [6], cats are still the ‘poor stepchild’ in companion animal medicine, receiving less attention in research into common medical problems or improved diagnostic and treatment approaches than is given to their canine counterparts.

Cats should not be considered or treated as small dogs as there are many differences. This review will be focused specifically on the cat mandible and other related aspects. Taking into account the key relevance of anatomy in high-quality medicine, the treatment of the feline mandible has many important features, most of them due to the singularity of its anatomy when compared to that of the dog. Knowledge of the functional anatomy of the cat mandible will give us clues as to appropriate treatment of the feline patient, minimising the occurrence of potential iatrogenic injuries. Mandibular injuries are varied: from those related to tooth extraction complications to mandibular fractures, or from temporomandibular joint pathologies to neural complications due to an excessive opening of the mouth (usually during oral or transoral manipulation). Consequently, we propose to review the cat mandible and the anatomically related structures from the point of view of functional anatomy with a clinical orientation, as it is of crucial importance in face morphology and for feeding and grooming. Thus, any alteration of the mandible would have important repercussions on the general appearance of the feline patient, affecting both food intake and grooming, as house cats can spend up to 50% of their waking time grooming their coat or performing related behaviour [5]. The various grooming behaviours are important to a normal healthy cat. In the absence of grooming, excess debris can tangle fur, causing painful tugging of the skin and even infection.

## 2. The Mandible

Starting with the anatomical description of the feline mandible, and according to the Veterinary Anatomical Nomenclature [7,8], the cat jaw has two halves (*mandibula*) joined rostrally by an *articulatio intermandibularis,* known as the mandibular symphysis (in carnivores, partially conformed by a *synchondrosis* and a *sutura intermandibulares*). Each mandibular half has a horizontal part (body, or *corpus mandibulae*) and a vertical part placed caudally (*ramus mandibulae*; Figure 1). The mandibular ramus has a dorsal part, oriented slightly caudally, with a coronoid process (*processus coronoideus*) on the top (which attaches the temporal muscle), and two faces with both *fossae* for muscle fixation: a masseteric fossa (very deep, laterally—*fossa masseterica*) and a pterigoid fossa (medially—*fossa pterygoidea*). More or less at the same level as the dorsal margin of the body, completely caudally, there is the condylar process (*processus condylaris*), a cylinder oriented horizontally that articulates with the mandibular fossa of the temporal bone and participates in the temporomandibular joint (TMJ) (*articulatio temporomandibularis*). On the caudal part and ventrally, there is an angular process (*processus angularis*), oriented caudally.

The mandibular body has two faces—medial or lingual (*facies lingualis*), and lateral or buccal and labial (*facies buccalis* and *labialis*)—separated by two margins ‒ the dorsal or alveolar (*margo alveolaris*), and the ventral—(*margo ventralis*). The ventral margin is smooth, whereas the alveolar margin is very irregular with deep pits where the root teeth are fitted (*alveoli dentales*). The body has two parts, a rostral or incisive part (*pars incisiva*) and the molar part (*pars molaris*). The incisive part supports the incisive and canine teeth, whereas the molar part contains the premolar and molar teeth. All the lower teeth form the *arcus mandibularis inferior*. The buccal face of the mandible is uneventful, except for the two mental foramina (*foramina mentalia)*: the main mental foramen and the posterior mental foramen (or three, the third, minute, one below the incisors, similar to a *foramen nutricium*, but through which the anterior mental nerves exit to innervate the area below the incisors) [9]. Lobprise & Dodd [10] also mention three foramina: the rostral foramen in the incisive part, the middle mental foramen at the level of the labial frenulum, and the caudal mental foramen between the two roots of the third premolar. In contrast, the lingual face has only a mandibular foramen (*foramen mandibulae*) in the rostral part of the ramus for the neurovascular supply of the mandible, which enters into the mandibular canal and provides the sole blood supply (the mandibular artery (*A. alveolaris inferior*‒ramus of the maxillary artery (*A. maxillaris*)—to the alveolar bone and teeth, and the mandibular vein (*V. alveolaris inferior*)) and the inferior alveolar (sensory) nerve (*N. alveolaris inferior).* According to Starkie & Stewart [11], a tough fibrous sheath surrounds the main nerve and its larger branches, and only the cat (in contrast to the rabbit and sheep) presents a sheath of compact bone surrounding the neurovascular bundle [11], a fact that in theory would help to localise the feline mandibular canal by radiography.

Pitakarnnop et al. [12] analysed 44 parameters on dried feline bones and demonstrated that there are only three hallmarks for sex identification in cats. One of them is the coronoid process of the mandibular ramus (with an accuracy rate up to 88.2%): ‘looking at a lateral view of the mandible, the coronoid process in females was more curved than in males’. The other two were described in the *os coxae* [12]. Hence, the mandible could give us additional and useful information for some scientific domains, such as forensic, developmental and evolutionary sciences, and also for zooarchaeological studies.

To the outer surface of the mandible is attached the Mm. *masseter* and *buccinator* (*partes buccalis* and *molaris*); the inner surface to the Mm. *pterygoidei* (*lateralis* and *medialis*), *mylohyoideus*, *geniohyoideus,* and *geniohyoglossus*; both surfaces at the lower border to the M. *digastricus*, and the upper process (*processus coronoideus*) to the M. *temporalis* [7,8].

When the jaw is closed, the lower incisors normally strike immediately caudal to the upper incisors. The lower canine occludes between the lateral upper incisor and the upper canine. Hence, this arrangement provides a shearing action, particularly between the cheek teeth [13]. The upper and lower teeth do not touch when the jaws move in a sagittal plane. However, when the cat chews on one side of the mouth, the lower jaw must be brought to that side, so the buccal surface of the lower teeth may shear upwards and forwards against the occlusal surface of the upper teeth [13]. 

Bite force is generated by the interaction of the masticatory muscles, the mandibles and maxillae, the TMJs, and the teeth. The main factors affecting the bite forces in dogs and cats are body weight and the skull’s morphology and size [14]. In cats, biting forces of 20–23.25 kg at the canine and up to 28 kg at the carnassial teeth have been reported [15] or, according to Kim et al. [14], an average of 73.3 Newtons (N) and 118.1 N, respectively. However, if converted to kg-force (kgf), it is obvious that these measurements do not match with [15], as they result in 7.47 and 12.04 kgf, respectively.

## 3. Temporomandibular Joint

It is worth pointing out that the temporomandibular joint (TMJ) is a cardinal feature that defines the class Mammalia and separates mammals from other vertebrates. Despite its status as a mammalian identifier, the TMJ shows remarkable morphological and functional variations in different species, reflecting the great adaptive diversification of mammals in feeding mechanisms [16]. During evolution, the common features of the TMJ (such as modified hinge joint, fibrocartilaginous articular surfaces, and two synovial joint compartments separated by an articular disc) persisted mostly invariable, except for a few species [17]. The simple components of the TMJ present adaptations, both in form and function, to satisfy the needs of the species, such as feeding and communication [17]. The evolutionary variants include adaptations in the orientation of the joint cavity from parasagittal (many rodents) to transverse (many carnivores), among other features [16]. However, function still remains a problem, because muscles, movements, and joint loads are to a great extent species-dependent [16].

The feline TMJ, which works as a hinge, is a synovial condylar joint formed between the condyloid process of the *ramus mandibulae* and the *fossa mandibularis* of the *pars squamosa* of the temporal bone. Caudally, the deep gutter of the temporomandibular articulation is bounded by a prominent retroarticular process placed behind the mandibular fossa [18]. Medial dissection shows a close relationship of the medial aspect of the articular capsule with the mandibular nerve, the tympanic cord, and maxillary artery. This area is particularly sensitive due to the exaggerated opening of the mouth when using different instruments or clinical exploration or dentistry [19].

It is the unique joint with a whole articular disc (*discus articularis*) that has developed owing to a slight articular surface incongruity [20]. However, in the cat, the temporomandibular joint is highly congruent [17] and the articular disc is very thin and poorly developed. According to Arredondo et al. [21], it is attached around its entire periphery to the capsule, dividing the synovial cavity into two separate spaces—dorsal and ventral. The periphery of the disc is irrigated by small branches from the articular temporomandibular artery [21]. In the cat, the condyloid process formed by the *caput mandibulae* consists of a slender, transverse roller 13–15 mm wide with a diameter of 2–3 mm (Figure 2). The axis of the articulated roller is oriented transversely at the line of the occlusal plane [17]. Caudally oriented, it presents a highly curved convexity medial to the mandibular body that narrows laterally and the outer end is often pointed [22].

The mandibular fossa, which lies under the base of the zygomatic arch and is 12–15 mm wide, has a concave *facies articularis* placed between the retroarticular process (*processus retoarticularis*), which is a caudoventral extension (medially placed) of the mandibular fossa, and a rounded and pronounced articular eminence (*tuberculum articularis*) rostrolateral to the mandibular fossa [22] (Figure 3). Therefore, in cats, the mandibular head is completely surrounded by bony structures of the temporal bone, with both eminences acting like two stops to limiting anteroposterior movements, but allowing mandibular motion to the sagittal plane of the cutting edge of the molar/premolar border P4/M1 (just opening and closing the mouth), with very limited lateral [23,24]. According to Crompton et al. [25], the retoarticular process resists the posteriorly directed force of the temporal muscle and the articular tubercle resists the anteriorly directed force of the superficial masseter muscle. When the mouth is closed, the mandibular dental arch fits into the dental arch of the maxilla and leaves no gaps to allow lateral or transverse movement. However, according to Knospe [22], the sideways movement of the lower jaw in the transverse plane occurs only when the oral cavity is slightly to strongly open, providing 2–3 mm to the right or left, with a total of approximately 5 mm, allowing crushing shears in the cat’s P4/M1, splitting the jaw pressure into a vertical cutting and a transverse pressure component.

Problems or perception of pain in opening or closing the mouth should always include a complete evaluation of the bilateral TMJ. Imaging with radiographs can be challenging due to superimposition of maxillofacial/cranial structures, requiring some rotation in either the lateral (10–30°) or long axis (10–30°) to isolate the individual structures [10].

In addition, Gracis and Zini [23] stated that the evaluation of the vertical mandibular range of motion or range of mandibular abduction (the distance between the maxillary and mandibular incisor teeth at maximum mandibular extension) should be incorporated into every diagnostic examination, as it may be valuable in showing changes over time for a single patient (concurrently with the patient under general anaesthesia and the musculature is relaxed). Consequently, early detection of a reduction in joint mobility allows a prompt diagnosis of these limiting pathologies or conditions. Those conditions may affect intra-articular or extra-articular TMJ structures, such as ankylosis secondary to fracture, joint luxation, dysplasia and osteoarthritis, which are relatively common TMJ lesions in cats [21], or fracture, osteomyelitis, bone neoplasia, retrobulbar masses, neuromuscular diseases, and trismus [23]. It should be taken into account that in cats there is a positive correlation between the vertical mandibular range of motion and (1) body weight and (2) age. In addition, male cats can open their mouth wider than can females [23].

## 4. Mandibular Teeth

The cat has double dentition: deciduous and later permanent teeth. It is edentulous at birth but develops a set of deciduous teeth that start erupting between 2 and 8 weeks after birth [13]. At 60 days, the deciduous dentition is complete [14]. Between 3 and 6 months of age, the deciduous teeth are shed as the permanent teeth erupt, and their full crown height should be achieved by 10–12 months of age [13]. There are four types of teeth depending on their shape and function: incisors (I), canines (C), premolars (PM), and molars (M). The permanent teeth number 30 in all. The formula of the permanent teeth includes I 3/3, C 1/1, PM 3/2, M1/1. The deciduous formula is similar but lacks molar teeth (26 teeth in total). The incisors and canine teeth of the cat are all single-rooted, as those of the dog. The mandibular first and second premolars are normally absent. The mandibular third and fourth premolars and the single mandibular molar each have two roots. The roots of the premolar are nearly equal in size, but the mesial root of the mandibular molar is approximately three times the width of the distal root [10]. The mandibular first molar tooth is considered a carnassial tooth (as well as the maxillary fourth premolar tooth).

Regarding the simple or brachydont tooth structure, as a general overview, the crown of the tooth is covered with enamel, whereas the roots are covered with cementum. Both hard-tissue layers meet at the cement–enamel junction, near the cervical portion of the tooth. Dentine constitutes the major part of the mature tooth. The difference between enamel and dentine microhardness is a result of the percentage of mineralisation they present. Dentine is synthesised by the odontoblasts lying at the pulp’s periphery. Since dentine is produced throughout life in a vital tooth, the permanent teeth of old cats have thicker dentineal walls and narrower pulp cavity compared to those of young cats [26]. The enamel thickness is reported to be 0.1–0.3 mm in cats and 0.1–0.6 mm in dogs [10] and, additionally, cat enamel is less hard than that of dogs, according to Hayashi & Hideo [27]. They also reported that enamel microhardness is higher in the outer layer than in the central or inner layer (since calcium ions in saliva infiltrate from the tooth surface), and there is an age-related increase in the microhardness of enamel [27]. In the premolars, the enamel hardness is higher at the top and middle of the crown, and it decreases in the cervical portion. Similarly, the dentine is harder in the cusp than in the rest of the tooth and the dentine microhardness decreases from the outer to the inner part. They also reported that the comparative microhardness enamel/dentine ratio varied from 3–9/1 [27]. Hence, data support that cat teeth are far more fragile compared to dog teeth; that being so, special care should be taken when manipulating.

The dental root is inserted in the dental alveolus and kept in place by the periodontium, which is made up of the gingiva, periodontal ligament, cementum, and alveolar bone. The space between the tooth and the free gingiva is the gingival sulcus, which should be no deeper than 0.5 mm in cats. The periodontal ligament attaches the root to the alveolar bone [26]. The alveolar bone appears with tooth eruption and disappears with tooth loss [14]. It surrounds the alveolar socket with an extension of cortical bone into the alveolus that outlines a radiopaque lamina dura in radiographic images [26]. The dental sac contacting with the cementum forms fibroblasts that produce collagen fibres while the other components of the periodontal ligament are developing. These are blood vessels, lymphatics, nerves, and various types of connective tissue cells. The nerves of the periodontal ligament are important as they provide additional senses to the tooth. They harbour pain fibres (similar to the pulp), but also pressure, heat, and cold fibres (not present in the pulp) [10].

However, as the aim of this review is not feline dentistry, specialised textbooks should be consulted for further information regarding teeth. Nonetheless, the more frequent dental pathologies seen in cats should be mentioned: periodontal disease and tooth resorption. Periodontal disease is perhaps the most common oral disease seen in dogs and cats and involves the periodontium and is the major cause of tooth loss. In general terms, periodontal disease is caused essentially by the accumulation of plaque on the tooth surfaces, and the severity correlates directly with the quantity of such deposits, producing gingivitis and forming a periodontal pocket. These changes lead to destruction of the gingival tissue with recession of the gums and retraction of the periodontal membrane. Ultimately there is infection of the dental root, destruction of alveolar bone, and dental loss. It is clear that a soft diet correlates positively with periodontal disease [13]. Cats suffer severely with progression of periodontitis and can be seriously anorexic. Gingival recession is common with marked tooth loss as a result of external tooth resorption, where the inflammation at the cement–enamel junction leads to collapse of the tooth crown with retention of the roots. These roots may become chronically inflamed or cystic and be very painful with periodontal abscessation. Treatment consists of removing degenerated teeth and any retained roots [13].

Dental disease is more common in older cats and can lead to other health problems, so maintaining oral health is important. To prevent dental disease, the single most effective method is to brush the cat’s teeth daily with a pet-specific toothpaste or powder, although it can sometimes take some time to train them to allow their teeth to be brushed.

It must be taken into account that the morphology of the mandible is conditioned by food habits [28]. As a carnivore, the cat has a mastication pattern consisting of an up and down or hinge movement of the mandible. Their TMJ lies on or close to the same plane as that of the lower dentition, with molars designed for crushing and slicing [25]. In contrast, in the herbivore group, which includes the Ungulates, the main action consists of a grinding movement of the mandible [28], as their TMJ lies above the occlusal plane of the maxillary dentition. In carnivores, the combination of a tall coronoid process and extension of the skull posterior to the TMJ permits both a large gape and a powerful bite. Another characteristic feature of this group is that the temporal muscle is considerably stronger than the masseter muscle [25].

## 5. Mandibular Neurovascular Supply

The first vessel to leave the maxillary artery in carnivores is the a. *articularis temporomandibularis* destined to the mandibular joint. The a. *alveolaris mandibularis* arises in a rostrolateral direction from the first part of the maxillary artery and runs towards the mandibular foramen through which it enters into the mandibular canal and provides blood supply to the mandible. Within this canal, it gives off the *rami dentales* to the molar and premolar teeth. Other branches pass through the alveolar canal to the canine and incisor teeth [29]. It exits at the caudal, middle, and rostral mental foramina to supply the lower lips [6]. In carnivores, the *rami mentales* leave the mandibular canal trough the mental foramina and ramify in the region of the *margo interalveolaris* in the gingiva of the incisor teeth and in the lower lip [29]. Veins often exist concurrently with arteries and empty by way of the maxillary and linguofacial veins into the external jugular vein [6].

In the mandibular canal, when examined in cross-section, the mandibular nerve is located in the dorsolateral portion of the canal with the vein in the ventromedial portion and the artery in the middle [10], although according to Davis & Story [30] the a. *alveolaris inferior* is situated lateral to the inferior alveolar nerve as the two enter the foramen.

The maxillary and mandibular branches (*nervi maxillaris* and *mandibularis*) of the trigeminal nerve (nervus trigeminus) are sensory, but the mandibular branch also supplies motor function to the masticatory musculature (temporal muscle, masseter muscle and the lateral and medial pterigoid muscles, which close the mandible) and other muscles. The digastric muscle is the only one responsible for opening the jaws; its rostral belly is stimulated by the mandibular branch of the trigeminal nerve, as the caudal belly is innervated by the facial (VII) nerve [10,31]. The facial nerve also provides motor function to many cutaneous facial muscles and is responsible for taste in the rostral two-thirds of the tongue [6]. The inferior alveolar nerve contains afferent fibres from the ipsilateral lower lip, areas of oral mucous membrane and mandibular teeth. It supplies sensory innervation of the lower teeth and, after exiting the mental foramina as the mental nerves, it innervates the lower lip. According to Robinson [9], fibres supplying the teeth are found in all branches, except the mental ones. At the mandibular foramen, the inferior alveolar nerve is a single bundle with the mandibular artery and vein lying inferior to it. Within the mandible, the nerve divides into several branches, which conform to a basic pattern with some individual variation. Three branches, splitting dorsally the main trunk, supply the alveolar processes (the posterior, middle, and anterior alveolar branches); another supplies the canine and incisor regions (canine/incisor branch), and there are four mental branches (posterior, main and two anterior) that leave through the various mental foramina. Interconnecting fibres are often seen between these principal branches [9]. There is no apparent bilateral symmetry and all the branches contain fibres from at least two adjacent tissues, including afferences from the pulp, periodontal ligament, mucous membrane, and skin. In addition, the nerves supplying one tooth come from different branches because they do not travel in a unique branch of the main trunk (Figure 4). The most proximal branch splitting from the superior aspect of the main trunk is the posterior alveolar branch. It contains afferents from the molar, fourth premolar, and occasionally third premolar teeth [9]. The next alveolar branch is the middle alveolar branch, leaving dorsally the main trunk beneath the distal root of the molar tooth to supply the third premolar, although in 50% of specimens it also supplies the canine teeth and the third incisor. The last alveolar branch is the anterior alveolar nerve, which supplies the canine and third incisor teeth in addition to mechanoreceptor afferents from the mucous membrane and skin adjacent to these teeth. Occasionally it also carries fibres from the second and first incisors and the third premolar teeth. Other branches splitting laterally from the main trunk, such as the posterior and main mental, leave the mandible through the posterior and main mental foramina, respectively, to supply the buccal gingival margin, the mucous membrane on the labial side of the alveolar process and the skin of the chin and lip from the anterior to caudal part (just up to the rostral edge of the molar tooth). The anterior mental branch divides in two terminal branches: the canine/incisor nerve and the anterior mental nerve. The latter does not contain any pulpal or periodontal afference. Lastly, the canine/incisor branch carries fibres from the canines and all three incisors.

Relative to the existence of transmedian innervation, Robinson [9] said that there is no evidence of transmedian innervation of tooth pulps; nonetheless, the cutaneous innervation in the anterior mental nerve crosses the midline for 1–2 mm. In contrast, Anderson & Pearl [32] reported the existence of an extensive transmedian innervation of the teeth in the cat; an innervation which is particularly dense in the canine teeth and extends at least as far laterally as the third premolar teeth. Wilson et al. [33], using the horseradish peroxidase technique, reported that, in addition to the inferior alveolar nerve, the nerve to mylohyoid and possibly other accessory neural pathways is involved in incisor innervation in cats.

According to Izumi et al. [34], ligation or cutting of the inferior alveolar nerve always elicits an increase in gingival blood flow. They also reported that blood flow in the cat gingiva and periodontal ligament is controlled by sympathetic α-adrenergic fibres for vasoconstriction; regarding vasodilatation, sensory fibres are involved besides the mast cells of the gingiva. According to Skerrit [31], a unilateral deficit of the mandibular nerve results in weakness of the chewing muscles when biting and atrophy of the temporal and masseter muscles. When the deficit is bilateral, it leads to drooping of the mandible and inability to close the mouth [31].

Mental nerve block, when needed, should be done at the main or middle mental foramen and anaesthetises the buccal soft tissues and the mandibular incisors and canine on the side injected [35]. The mandibular nerve can be anaesthetised by intraoral or extraoral techniques. Anaesthesia of the nerve results in desensitisation of the mandibular body, the lower portion of the mandibular ramus, all mandibular teeth on the same side, the labial/buccal surfaces of the mandible, and the mucosa and skin of the lower lip and chin [35].

The mandibular canal is not a medullary canal, and treating fractures of the body via an intramedullary pin through this canal will damage the associated neurovascular bundle. In many fractures or tumoural reparative surgery of the mandible, the inferior alveolar nerve is damaged and then resected. However, some experiments provide evidence that peripheral nerve fibres are important, not only in normal bone homeostasis and skeletal growth, but also in their influence on the repair mechanism of bone fracture. Many experiments suggest that sensory and sympathetic nerve fibres do have a role in bone remodelling and osteogenic differentiation of precursor cells during skeletal growth. Hence, the loss of sensory nerves could result in a decrease in the quality of new bone, as reported by Cao et al. [36] on the mandibular distraction osteogenesis in rabbits, stating that the peripheral sensory nervous system plays an important role in bone regeneration. Sensory nerves also play an important role in regulating bone resorptive activity, as shown by Yamashiro et al. [37] during experimental tooth movement in rats. In bone, the areas with the highest metabolic activity receive the richest sensory and sympathetic innervation, which has an effect on the activity of both osteoblasts and osteoclasts [37]. As seen in many articles, the inferior alveolar nerve could also be affected by reparative osteosynthesis, and as in the feline mandible there is very little space to safely place the screws required to fix the metal plates to resolve the fracture. Regarding the integrity of the inferior alveolar nerve, we consider a conservative option should be prevalent over inferior alveolar nerve resection so as to not impair the outcome of bone repair and sensitivity. Hence, we propose that surgeons should be as conservative as possible and try to leave the nerve intact, with no attempt to pull the nerve as this causes a lesion, and being sure where to place the screw in order to not affect the neurovascular bundle or any dental root.

## 6. Mandibular Radiographic Images

Dental radiography requires general anaesthesia to get accurate projections and avoid any trauma or damage to the equipment.

Cats have two mandibular halves that connect rostrally through a mandibular symphysis. The symphysis is represented radiographically as a radiolucent border between the two mandibles. The portions of the mandibles associated with the symphysis are roughly parallel [38].

The lateral projection of the feline mandible (Figure 5) is bordered ventrally by the ventral cortex and dorsally by the cusps of the premolars and molar. The area corresponding to the location of the mandibular canal, containing the neurovascular bundle (the alveolar mandibular nerve, artery and vein) appears as a radiolucent area just dorsal to the ventral cortex and ventral to the dental roots. When evaluating any tooth, the following are assessed: crown (and enamel) and pulp chamber, root and root canal, periodontal ligament space and alveolar bone. The relative radiolucent line outlining the roots (*lamina lucida*) is the periodontal ligament space. It is wider early in age and is typically widest at the coronal and apical one-third of the root [38]. Adjacent to this ligament space is a radiodense line (*lamina dura*), which is the cortical bone of the alveolus. Contiguous to it is the trabecular bone of the alveolus. The crown is covered by a more radiodense margin, which is the enamel, and the bulk is dentine, which is not as radiodense as enamel but is radiodense compared to bone. Because the cementum has nearly the same radiodensity as bone, it is not obvious radiographically. In multirooted teeth, bone should be present up to the apex of the furcation. The centre of the root is the radiolucent root canal, which houses the radicular portion of the pulp.

According to Milella & Smithson [39] ten radiographic films should be taken to assess accurately each tooth in the cat’s mouth (upper incisors, upper left canine (anterior-posterior oblique and lateral), upper right canine (anterior-posterior oblique and lateral), upper left maxillary premolars and molar, upper right premolars and molar, lower canines and incisors, lower right mandibular teeth, lower left mandibular premolars and molar). However, in older cats, full mouth radiographs are more to be recommended.

Hoffman & Ridinger [40] pointed out the value of obtaining annual dental radiographies, as well as the importance of interpreting dental radiographic findings in the light of the patient’s systemic health and oral examination findings. This information will be essential when planning treatment in differentiating between primary oral disease and those secondary to systemic disease.

## 7. Topographical Considerations

The arteries of significance in clinical dentistry and oral surgery have the same origin in the common carotid artery (*arteria carotis comunis*), which branches into the internal and external carotid arteries (*arteriae carotis externa* and *interna*) [10]. The internal carotid artery blood supply is insignificant in the cat. Consequently, the external carotid artery, continuing as the maxillary artery, provides the majority of cerebral blood flow in the cat. The maxillary artery lies medial to the angular process of the mandible and branches into the maxillary rete before entering the skull through the orbital fissure [10]. Exclusively in the cat, the maxillary artery forms a network known as *rete mirabile a. maxillaris*, which is extracranial, near the *foramen ovale*. This network extends dorsally and laterally to the apex of the periorbital region. In the cat, the arteries for the eye and accessory structures arise from this network as do also the *rami retis*, which pass through the *fissura orbitalis* to connect with the *circulus arteriosus cerebri*. At the same time, the maxillary artery itself, being traceable through this network as a stronger vessel, leads the *a. infraorbitalis* [28].

According to Skerritt [41], there is an inverse relationship between the degree of development of the internal carotid and that of the anastomosing ramus of the maxillary artery: when one is large, the other is small. In no species are both of these channels fully developed. In cat and sheep, the lumen of the internal carotid artery becomes obliterated in the weeks or months after birth (although at birth it is fully functional). As a result, the whole of the adult brain is supplied by maxillary blood via the anastomosing ramus of the maxillary artery. In addition, a *rete mirabile a. maxillaris* occurs on the anastomosing ramus of the maxillary artery in all species in which the supply from the maxillary artery is well developed [41].

The significance of the *rete mirabile* has long been discussed. It was thought that it might eliminate pulsation before the blood reaches the brain itself. However, more recent observations indicate that the rete is involved in thermoregulation, as reported by Baker & Hayward [42]; using different anatomical nomenclature, they stated that the plexus of arteries that make up the carotid rete in the cat seems to be able to modify the temperature of central arterial blood as it enters the cranial cavity. In this short paper (only three pages) published in Nature, they described that in the cat, the carotid rete lies outside the cranial cavity, near the apex of the orbit, and is termed an ‘extracranial rete’, while in artiodactyls the rete is intracranial, lying in the cavernous sinus at the base of the skull. The extracranial rete lies within a venous lake, and the large surface area of the interlacing network of vessels may permit a ‘countercurrent’ exchange of heat between the arterial blood of the *rete* and the venous return from various regions of the head, having a profound effect on brain temperature that may have significant thermoregulatory consequences, allowing large, rapid changes in the temperature throughout the cranium (0.1–0.7 °C). The presence of the extracranial carotid rete is the most important cerebrovascular difference between the cat and the other species studied by Baker & Hayward [42].

Consequently, in the cat, maxillary arterial blood is distributed to all of the brain and any disturbance of the blood flow may have dramatic consequences. Regarding this aspect, it is currently proven that overextension of the mandibles of the cat (during oral and transoral procedures, such as intubation) can lead to compression of the rete and/or compression of the maxillary artery by the angular process of the mandible (Figure 6), leading to cerebral ischaemia and resulting in temporary or permanent cortical blindness, loss of hearing, or possibly death [43,44].

A comprehensive understanding of the functional anatomy of all structures associated with the caudal angle of the cat mandible may explain why keeping a cat’s mouth wide open for a prolonged period of time may result in temporary or permanent neurological deficits, unilaterally or bilaterally, post-anaesthesia [26]. Cats and dogs have the maxillary artery running through this area, but what gives only cats an increased risk of cerebral ischaemia when the mouth is wide open is that the mandibular angular process, presses against the area through which the maxillary artery passes. This reduces to some extent the maxillary artery blood flow. However, an additional feature is decisive to this occurrence: the internal carotid artery (leading the main blood supply to the brain, retina and inner ear) is functionally absent in cats, so all the blood to the brain is supplied exclusively by the bilateral maxillary arteries. Consequently, the longer the duration of the pressure, the higher the risk of onset of cerebral ischaemia and/or blindness and/or deafness. De Miguel García et al. [44] suggested (where the use of a mouth gag is essential for surgery) to reduce the size of the gag or to close the cat’s mouth every few minutes (throughout the procedure) to enable restoration of the blood supply. However, the reason why cats are unequally affected is still unknown, although it could be due to collateral or altered blood flow through the basilar arteries (although, according to Skerritt [41], they only carry blood away from the arterial circle).

Thus, the use of spring-loaded mouth gags is no longer recommended in feline patients [10,26] as they apply continual force to keep the mouth open to such an abnormal degree [45]. It is also reported that the vascular flow is more compromised on the side ipsilateral to the mouth gag, perhaps because the distance between the angular process of the mandible and tympanic bulla is smaller on the ipsilateral side [46]. Another drawback would be that the tighter the lips and cheek become when the mouth is wide open, the more difficult it is to retract them, to perform surgery or a thorough oral examination [26]. In contrast, custom-made plastic mouth props (made from a syringe cap) have many benefits: (1) they are gentler on the jaws and appear to induce fewer alterations in blood flow; and (2) they are radiolucent, hence they do not interfere with diagnostic radiological imaging [26], among others. However, if a mouth gag must be used in a feline patient, the smallest possible gag (ideally between 20 and 30 mm), should be chosen to retract the upper and lower lips without difficulty and the duration of use minimised [26,47].

The Nomina Anatomica Veterinaria [7] does not clarify which species present molar salivary glands (*glandulae molares).* Okuda et al. [48] described the bulge lingual to the lower molar tooth to be a small salivary gland corresponding to the lingual molar gland. Interestingly, there is no equivalent salivary gland in the dog. Previously, Orsini & Hennet [14] stated in their review entitled ‘Anatomy of the mouth and teeth of the cat’ that ‘just medial to the lower first molar on the floor of the oral cavity is a mass-like flap of oral mucosa with no known function; its prominent appearance leads to an incorrect identification as an abnormal finding’. This major salivary gland in the cat has two parts (Figure 7): the buccal and lingual molar gland. The secretory portion of the buccal molar gland is close to the commissure (between the M. *orbicularis oris* and the mucous membrane of the lower lip at the angle of the mouth) and it empties into the buccal cavity by several small ducts; and the lingual molar gland located within a membranous bulge caudolingual to the mandibular molar tooth (constituting the molar pad) [10]. Sometimes, the membranous molar pad enlarges and may be traumatised (when chewing), being important not to be mistaken for a tumour or polyp [49].

## 8. Conclusions

Clinicians must have a deep knowledge of the functional anatomy of cats (taking into account that they present some important differences with respect to the dog) to achieve an effective and high-quality cat medicine.

A good knowledge of the anatomy of the mandible and the TMJ, and their relation to other important structures, such as blood vessels and nerves, is essential for an accurate interpretation of radiographic images and tomographic diagnostic techniques, in order to make a diagnosis and achieve good results in the management of different conditions. 

Clinicians and surgeons are increasingly aware of animal welfare to avoid or, at least, minimise any suffering due to iatrogenic complications. Such as those related to a temporary or permanent blindness and/or deafness following general anesthesia when the mouth is held open with a gag (to the compression of the angular process on the maxillary artery, disrupting the blood flow to the maxillar *rete mirabile* and the brain); the resection of the lingual molar pad (that includes the lingual molar salivary gland), or those complications secondary to dental extraction.

## Figures and Tables

**Figure 1 animals-11-00405-f001:**
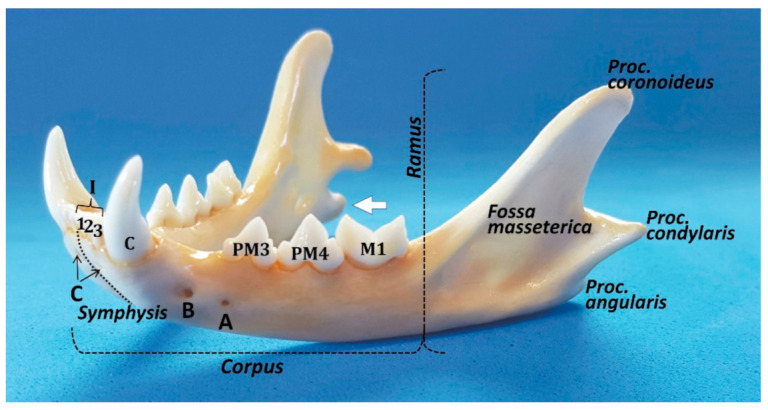
Rostro-lateral view of the cat mandible. The most relevant details are shown. A: posterior mental foramen; B: main or middle mental foramen; and C: rostral mental foramen. The prominence of the *processus angularis* is more manifest in a medial view (white arrow).

**Figure 2 animals-11-00405-f002:**
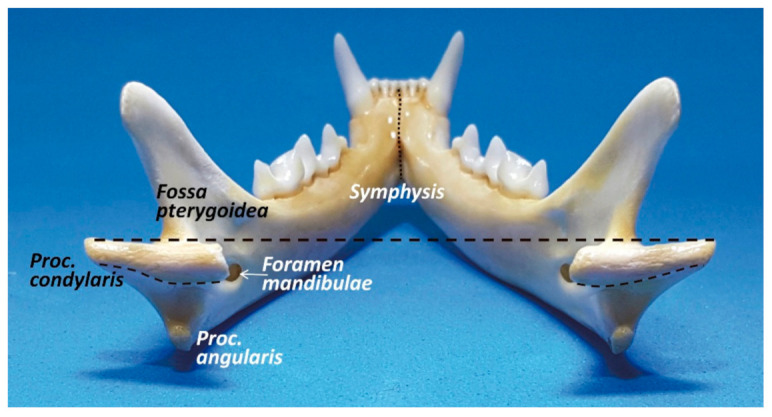
Caudal view of the cat mandible. Both cone-shaped condylar processes are horizontally oriented.

**Figure 3 animals-11-00405-f003:**
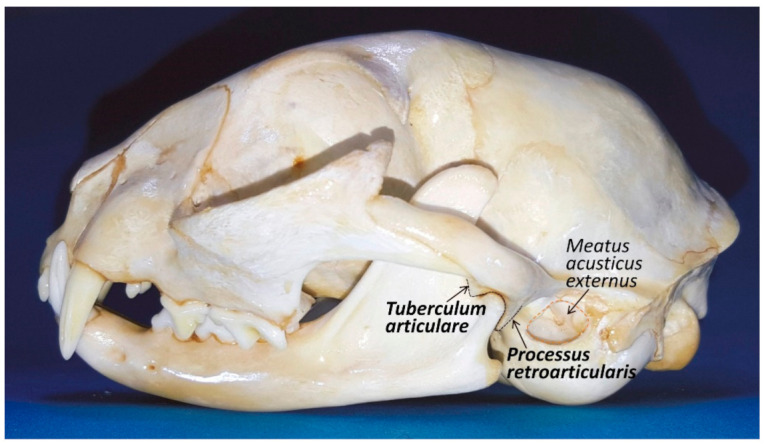
Lateral view of the cat head. The temporomandibular joint has two ‘bumpers’ (one rostrally and other caudally) to keep in place the condylar process in the *fossa mandibularis* of the temporal bone.

**Figure 4 animals-11-00405-f004:**
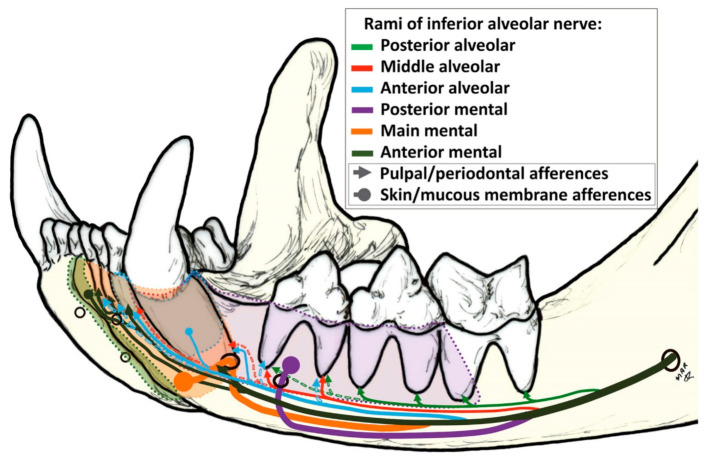
Innervation of the cat mandible modified from Robinson [8]. Diagram showing the inferior alveolar nerve through the mandibular canal and its division in *rami* to innervate different structures. Dashed lines indicate inconstant afferences. The thickness of the line when detaching from the inferior alveolar nerve is proportional to its real thickness, according to Robinson [8].

**Figure 5 animals-11-00405-f005:**
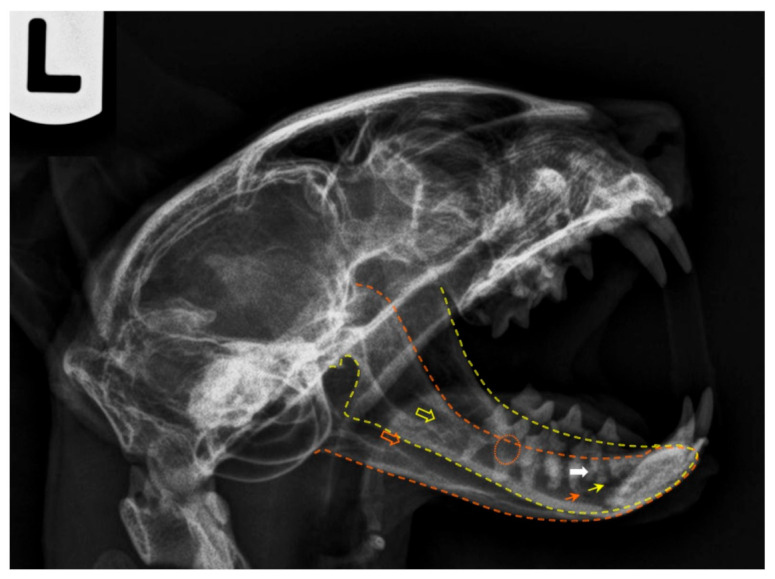
Postmortem radiography of a cat head, laterolateral projection (left lateral recumbency). The yellow outline corresponds to the left mandible and the orange one to the right jaw. The respective coloured empty arrows point the mandibular foramina, and the small arrows indicate the main mental foramina. The orange circle shows signs of tooth root resorption and the white arrow points evidences of a dental root fracture. Note that a plastic needle cap was used to keep the mouth open. As plastic is radiolucent, it does not interfere with the radiologic image.

**Figure 6 animals-11-00405-f006:**
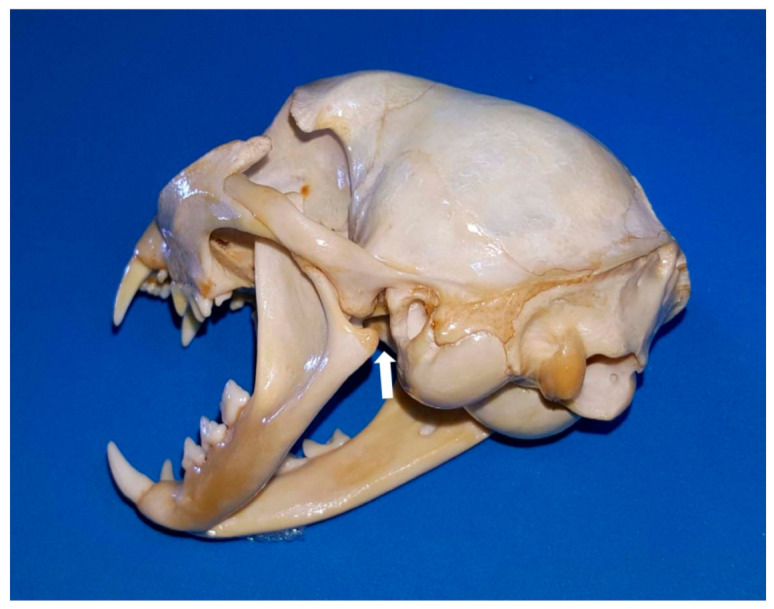
Caudolateral view of the cat head with the mouth open. The white arrow points to the channel narrowed by the *processus angularis*, through which the *a. maxillaris* runs to lead the *rete mirabile.*

**Figure 7 animals-11-00405-f007:**
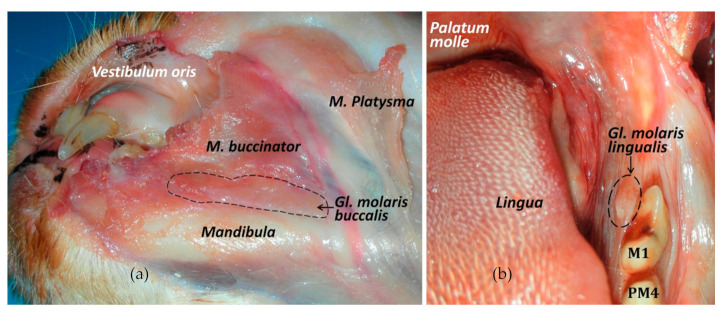
The cat molar salivary glands. (**a**) Ventrolateral view of the buccal molar gland. (**b**) Frontal view of the lingual molar gland, caudomedially to the first molar tooth.

## Data Availability

Data sharing is not applicable to this article as no new data were created or analyzed in this study.

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
