# Peer review of "The Cat Mandible (I): Anatomical Basis to Avoid Iatrogenic Damage in Veterinary Clinical Practice"

_animals, 2021, doi:10.3390/ani11020405_

Round 1

Reviewer 1 Report

Reviewers’ Comments and Authors Response

Paper number: animals-1065165

Paper title: “The cat mandible (I): anatomical basis to avoid iatrogenic damage in veterinary clinical practice”.

The manuscript quantifies the cranial morphology of a large data set of pet rabbits.

The paper is well written, providing useful information with a very complete anatomical description. Sections are well organized. Similarity is only 3%.

This paper has a potential to be accepted. I only would suggest in Ln: 134 to add something loke “and also for zooarchaeological studies”

In my opinion, the paper can be considered for publication.

Reviewer 2 Report

The manuscript "The cat mandible (I): anatomical basis to avoid iatrogenic damage in veterinary clinical practice" summarizes the anatomical knowledge of the cat mandible in clinical aspect. The subject of this review manuscript is current especially in veterinary practice.

However before the acceptance this manuscript needs revision.

Suggestion for Authors:

  1. All the anatomical terms should be used according to "Nomina Anatomica Veterinaria", Sixth edition - 2017 - please add this additional citation to the reference list.
  2. Please check all the anatomical terms - these terms should be unified (the terms in Latin or in English - because now these terms are mixed in several points. For example within the abstract the Authors used: "the inferior alveolar arteria and vena" change as: "the inferior alveolar artery and vein".
  3. on Figure 1 and Figure 2: "symphisis" change as "symphysis"
  4. on Figure 3: "meato acusticus externus" change as: "meatus acusticus externus"
  5. on Figure 7: "pallatum molle" change as: "palatum molle"

Reviewer 3 Report

This manuscript reads like an introductory comparative anatomy student laboratory manual. Any practitioner who might otherwise benefit from the manuscript will already be eminently familiar with all the particulars of basic anatomy described in unnecessary detail that apply equally to virtually all mammals. At the very least, the excessively verbose manuscript could be slashed to a fraction of it’s length by simply using accepted anatomical terminology and citing other references if deemed necessary. Many entire paragraphs could be deleted without loss of substance (e.g., lines 88-127, 135-138, 263-275, 313-322, the bulk of 415-433, just to name a few). Others could be shortened as single direct statements (e.g., condense lines 236-241 to the deciduous and permanent dental formulae). Figures 1-3 are similarly superfluous. There is a great deal of text that is simply tangential and irrelevant (e.g., what is the purpose of reporting enamel thickness in humans, whose tooth morphology, numbers, absolute size, and chewing mechanics differ utterly from those of cats and dogs). Correct anatomical terminology is not uniformly used (e.g., line 222, “vertical mandibular range of motion (the distance between the maxillary and mandibular incisor teeth at maximum mandibular extension)” should be concisely stated “range of mandibular abduction”; line 255, What is "inside" in anatomical terminology? How about simply saying “calcium ions in saliva infiltrate from the surface”; line 411, cats do not “have paired mandibles”, they have paired dentaries which make up the mandible and is also redundant with lines 89-90 which are themselves incorrect – fibrocartilage joints are by definition syndesmoses [defined by protein species], not synchondroses, except in those special cases in which they are both). Other redundancies are numerous (e.g. lines 247-249 vis-a-vis 432-433). I tire of citing examples.  Not much of substance begins before page 8 of 16. I question why this manuscript could not be drastically condensed,  abbreviated and absorbed into the companion manuscript (which I declined to review and therefore have not read).

Although the manuscript appears to be free of misspellings, there are a few awkward or grammatically incorrect sentences (e.g., line 69, change “differential aspects” to “differences”). What is meant by “sensible” on line 180? Unrelated concepts are bundled together in single paragraphs. The paragraph from lines 150-163 is poorly written for this and other reasons. It begins with a bizarre discussion of relative mandibular size and concludes with an equally confused discussion of bite force. What is meant by “smaller”? It is not physically possible for the mandible to be “smaller” than the maxilla in longitudinal linear dimension. Moreover, the mandibular dental arcade is and remains narrower than that of the maxilla in both dogs and cats throughout life. It has to be in order for the carnassial teeth to function (i.e., the wear facet of P4 occludes lateral to m1), and this much is even acknowledged. So what are the authors describing? It is ironic that the "mandibular arch" is not defined in their excessive description of mandibular morphology. What is meant by “when the jaws move in a vertical line”? In what plane is this vertical line? What is the “palatal” surface of a tooth – lingual? Of course the bite forces of canines and carnassials don't match. The canines have longer moment arm, well anterior to the masseter whereas the origin of the masseter is equally distant from the TMJ as are the carnassials. Neither references describing the comparative anatomy and functional significance of the retroarticular process nor the comparative mechanics of chewing in mammals are cited. Statements about the orientation of the mandibular condyles (line 189 and elsewhere) should be corroborated with measurements derived from 3-dimensional CT imagery. More important is the degree to which the condyles are not oriented in the same axis of rotation throughout mandibular ab/adduction.

I recommend eliminating nearly all that isn’t specific to cat, or that describes useful (to clinicians) information about how the cat is fundamentally different from other relevant species. Some of it appears to be simply wrong (e.g., the intrarticular disc of the TMJ separates two distinct synovial cavities even in humans). I recommend condensing the excessively verbose, conversational, and redundant descriptions of those details that are relevant to the cat (e.g, lines 491-544). I emphasize that in a potential revision of this manuscript, my examples should not be considered comprehensive. A more concise presentation would be more to the point and convey the message more effectively. At the risk of my own redundancy, I recommend that the important parts of this manuscript be rewritten concisely and incorporated into the companion manuscript.

Round 2

Reviewer 2 Report

Suggestions for Authors:

The resolution of Figures 1, 2, 3, 4 and 7 is too low, please make the correction of these figures (words within the figures are weakly visible)

Reviewer 3 Report

The authors have at least addressed most of my stated concerns, albeit that I asserted my comments were examples and not intended to be comprehensive. The resubmission is far from being a major revision, so of course it is far from meeting my original recommendation. Nevertheless, the manuscript is much improved. I still think that much could be abbreviated assuming that readers have at least perfunctory formal training in anatomy.

The authors missed my point that “Unrelated concepts are bundled together in single paragraphs” to which they responded “We must insist on the usefulness of having all the anatomical information about the cat mandible (or at least most of it) compiled in a single document”. My statement concerned the grouping of unrelated topics into a single paragraph, not whether the topics should all be included in the manuscript. Concepts are most effectively communicated when each paragraph addresses a single topic that is introduced by a leading topic sentence, and concluded with a summary statement that emphasizes the importance of the topic and leads to the next related paragraph. This is simply an issue of good writing.

Gratuitous references to comparative and evolutionary studies are a good example of why it is important to adhere to universally accepted anatomical nomenclature. For example, it is of no use, in fact it is patently confusing, to speak of paired mandibles or ‘jaws’ if this is intended to be intelligible to evolutionary biologists, regardless of whether this peculiar verbiage is uniquely meaningful to anyone else. It is anatomically incorrect to speak of paired mandibles or jaws; this much is implicitly acknowledged in the author’s own response to my review in which they write  “Dyce: “In the cat, the halves of the mandible do not fuse…””. Why don’t the authors simply refer to the two halves, instead of incorrectly calling them mandibles or jaws? Similarly, there are two pairs of terms that describe the ‘opening and closing’ of the ‘mouth’, i.e., adduction/abduction and elevation/depression. This is not something worthy of quibbling over, either by me or by the authors.
